# Single-Line LiDAR Localization via Contribution Sampling and Map Update Technology

**DOI:** 10.3390/s24123927

**Published:** 2024-06-17

**Authors:** Xiaoxu Jiang, David K. Yang, Zhenyu Tian, Gang Liu, Mingquan Lu

**Affiliations:** 1Lab for High Technology, Tsinghua University, Beijing 100084, China; jxx_315@163.com (X.J.); zero_tin@163.com (Z.T.); 2College of Arts and Sciences, University of Washington, 1400 NE Campus Pkwy, Seattle, WA 98105, USA; dayang02@uw.edu; 3Department of Electronic Engineering, Tsinghua University, Beijing 100084, China; lumq@tsinghua.edu.cn

**Keywords:** localization, single-line LiDAR, robustness, degenerate environments, contribution sampling, map updating

## Abstract

Localization based on single-line lidar is widely used in various robotics applications, such as warehousing, service, transit, and construction, due to its high accuracy, cost-effectiveness, and minimal computational requirements. However, challenges such as LiDAR degeneration and frequent map changes persist in hindering its broader adoption. To address these challenges, we introduce the Contribution Sampling and Map-Updating Localization (CSMUL) algorithm, which incorporates weighted contribution sampling and dynamic map-updating methods for robustness enhancement. The weighted contribution sampling method assigns weights to each map point based on the constraints within degenerate environments, significantly improving localization robustness under such conditions. Concurrently, the algorithm detects and updates anomalies in the map in real time, addressing issues related to localization drift and failure when the map changes. The experimental results from real-world deployments demonstrate that our CSMUL algorithm achieves enhanced robustness and superior accuracy in both degenerate scenarios and dynamic map conditions. Additionally, it facilitates real-time map adjustments and ensures continuous positioning, catering to the needs of dynamic environments.

## 1. Introduction

Real-time localization is crucial for autonomous robot navigation, especially in indoor environments where global positioning signals, such as GPS (Global Positioning System), are unavailable. Researchers have explored various sensors to achieve reliable positioning. Cameras, for instance, achieve high precision by capturing images for feature matching with stored maps; however, their performance diminishes with changes in lighting and perspective. While odometry is less susceptible to external influences, it suffers from error accumulation. Additionally, localization via environment-embedded wireless beacons—utilizing signals such as RSSI (Received Signal Strength Indicator), TOF (Time of Flight), TDOA (Time Difference of Arrival), and AOA (Angle of Arrival)—is hindered by beacon placement prerequisites and environmental factors, such as obstructions and multipath effects.

Advancements in autonomous driving and robotics have established LiDAR as a premier sensor choice for localization and navigation [1]. LiDAR can be separated into two primary categories: multi-line and single-line. Multi-line LiDAR provides 3D scanning, while single-line LiDAR focuses on planar distances and positions across horizontal planes. Due to its sophistication, cost efficiency, minimal energy requirements, and swift refresh rates, single-line LiDAR is well suited for indoor robots, ranging from floor sweepers to logistics bots.

Single-line LiDAR positioning aligns scan frames with preexisting maps [2], ensuring centimeter-level accuracy via particle filters or graph optimizations. Its benefits include its exceptional accuracy, minimal computation, and straightforward setup. Despite these advantages, LiDAR’s robustness is challenged in industrial and domestic applications. Narrow passages and constrained areas can lead to a significant focus of LiDAR point clouds in limited directions, obscuring other vectors and resulting in drift or complete localization failure, thus endangering the success of subsequent navigation tasks.

The dependency on static maps for matching introduces another layer of vulnerability, as environmental alterations—such as vehicle movements or material handling—disturb map fidelity, destabilizing the backend matching algorithm.

To address the robustness issues associated with single-line LiDAR localization, we present a framework using contribution sampling in degeneration scenarios—environments where point cloud data are unidimensional. By recognizing and filtering potential impact points within the alignment’s degeneration state, we not only achieve high-precision positioning under complex conditions but also eliminate drift. Furthermore, we develop a map anomaly detection and real-time update mechanism to minimize the detrimental effects of environmental changes on localization.

The primary contributions of our work are the following:Introducing a robust single-line LiDAR positioning framework with advanced optimization techniques for complex environments.Proposing a contribution sampling method in degeneration scenarios to enhance registration and localization stability.Developing a detection and update algorithm for maps, integrating ray casting and a multi-perspective weighted voting scheme, enabling prompt updates, and increasing the positioning algorithm’s adaptability to environmental changes.Conducting extensive evaluations of our system across a variety of challenging conditions, experimentally substantiating our algorithm’s effectiveness.

## 2. Related Works

High-precision LiDAR localization is crucial for autonomous navigation among mobile robots, with the particle filter-based Monte Carlo algorithm being a primary method. A novel concept of the “region of no concern” was introduced by [3], utilizing a heuristic weight function within the data fusion process of a 360° laser scan. Subsequent advancements, including multi-sensor Monte Carlo Localization (MCL), enhance accuracy through the use of weighted samples that quantify the robot’s positioning certainty with updates provided by laser rangefinder sensors [1,4,5]. An adaptive approach to MCL was explored by [6], proposing an algorithm that merges laser scan data with adaptive techniques to enhance real-time position and orientation estimates. In addition, visual recognition was integrated into MCL by [7] to improve localization in dense, non-planar settings, highlighting the challenges faced by traditional 2D LRF applications in such environments. Adaptive Monte Carlo Localization (AMCL) [8] offers an evolution of the MCL algorithm, showcasing flexibility in response to environmental variations by adjusting its sampling strategy based on sensory and motion data. A localization algorithm based on corners and line features using a single-line laser scanner is proposed [9]. This algorithm employs a weighted Iterative Closest Point (ICP) method for matching to attenuate the computational effort while preserving localization robustness. However, challenges for single-line LiDAR localization include localization drift, low confidence levels, and a lack of robustness in long corridors or dynamic spaces prone to feature loss.

Degeneration is one of the significant challenges that affect the robustness of LiDAR localization. This issue deeply affects the performance of robot position estimation. Degeneration, also interpreted as limitations of LiDAR measurements under certain conditions, triggers the failure of localization algorithms. A common instance includes inaccurate laser measurements due to the distribution of environmental features, such as the profound drift in long corridor environments. Several studies [10,11] have presented models to estimate the variance of position. Yet, the diverse nature and origins of errors complicate the precise modeling of these errors. Optimization-based state estimation methods to solve these degeneration issues were discussed by Zhang, Kaess, and Singh in 2016 [12]. Their suggested methods mitigate degeneration by analyzing the geometrical structure of the scenario. They highlighted the need for understanding problem constraints in devising effective solutions. Despite this, their approach does not fully utilize the available data under degraded conditions. In 2019, Zhen et al. [13] released LiDAR and UWB for robust localization in tunnel-like environments, but this solution increases sensor and deployment costs. In 2020, Hinduja et al. [14] implemented Zhang’s degeneration detection method in SLAM and updated it to well-constrained directions in the state space. While their mechanism fused the degeneration detection in the SLAM framework, the underlying issue in the ICP remains. In 2021, Dong et al. [15] introduced a PCA-based segmentation of point clouds to deal with feature scarcity and degeneration in robotic SLAM systems. Yet, its lack of broader applicability limits its effectiveness. In 2021, Nashed et al. [16] also proposed penalizing motion variations along the degeneration direction, but this method increases the computational complexity and does not directly improve the optimization for ICP. Ref. [17] employed a single 2D LiDAR and developed a grid-matching algorithm grounded in data association to achieve robust matching in the presence of significant noise, although the robustness enhancement in degenerate scenarios was not explicitly addressed.

The dynamic nature of map environments significantly influences laser localization robustness. Multiple studies have incorporated methods to maintain accurate localization despite environmental fluctuations. Aijazi et al. (2013) [18] developed a method to update laser maps by removing transitory objects through preprocessing segmentation and refining maps with point cloud matching. Despite advancements, this approach poses data processing complexities and considers storing and computing demands in large-scale applications. The removal of temporary objects also introduces a subjective element that could reduce change detection precision. Continuing this line of research, Schlichting et al. (2016) [19] devised an algorithm that discerns between static and dynamic objects using a tree classification technique for vehicle localization. However, this method incorrectly identifies changes and struggles with trees adjacent to larger structures at times. In a different approach, Withers et al. (2017) [20] attempted to quantify the error distribution for each point in a map’s point cloud, enhancing localization across several instances. Despite these efforts, constructing error histograms proved to be a complex task that requires further examination and refinement. Ding et al. (2018) [21] introduced a visual–inertial framework that combines point clouds, laser, and vision data. This requires the use of additional visual sensors and involves complex equations for detecting anomalies, which may prompt concerns regarding computational efficiency and practical application. Ref. [22] addresses artifact issues with a plane-based filter on ray-casting, but it faces challenges such as computational intensity, stringent accuracy demands for plane extraction, and difficulty in processing complex scenes. Ref. [23] studied a crowd-sourcing framework based on probabilistic models, where map changes are detected using probabilistic and evidential theory considering LiDAR characteristics. However, this framework cannot operate independently on a single vehicle and requires collaboration among multiple vehicles to function effectively. Ref. [24] studied the impact of indoor specular reflections on LiDAR and proposed a density-based clustering approach to distinguish between reflective and non-reflective objects. However, this method only addresses specular reflections and does not effectively account for changes in the map itself. To address the map update issue for multiple robots, Ref. [25] proposed a global grid fusion strategy based on Bayesian inference, which is independent of the order of merging. This method requires multiple vehicles to operate collaboratively and does not solve map change problems encountered by a single vehicle.

These discussions underscore the ongoing challenge within robotics research of developing reliable localization solutions that can effectively manage map changes and optimize for degeneration. This paper contributes to the field by proposing a unified approach that addresses both the degeneration in degraded scenes and the monitoring of map alterations, thereby striving to improve the robustness of the LiDAR localization system.

## 3. Materials and Methods

The proposed CSMUL algorithm framework comprises two essential components, as depicted in Figure 1: a front-end robust localization module and a back-end map-monitoring module.

The front-end module begins with initial localization via vehicle odometer readings or IMU-derived position information. Utilizing LiDAR scans and existing map data, this module actively identifies scene degeneration. The subsequent contribution sampling sub-module assigns weights to individual points within the point cloud, enhancing the localization process. Leveraging the point-to-line (PL-ICP) matching algorithm, the front-end module delivers an enhanced position estimation.

Concurrently, the back-end module refines the map with ray-casting and a multi-view weighted voting system, ensuring that the map remains current. By constantly integrating the updated map into the front-end module, this system reinforces the precision and stability of position estimations.

### 3.1. Front-End Robust Localization Module

#### 3.1.1. Degeneration Detection

Inspired by [12], we delve into the intrinsic mechanism of degradation in research scenarios within optimization equations. Upon the arrival of a new point cloud frame, we use previous localization results combined with pre-integration IMU to recursively provide initial values. An optimization equation for point-to-line matching is established based on the prior map.
(1)$$\underset{\theta ,t}{\arg\min},\sum_{i=1}^{N}∥{\vec{n}}_{i}^{T}\left[R\left(\theta \right)p_{i}^{s}+t-p_{i}^{k}\right]∥$$

Here, $p_{i}^{s}$ represents a point in the source point cloud (scan frame point cloud), while $p_{i}^{k}$ and ${\vec{n}}_{i}^{T}$ denote a point and the local normal vector in the target point cloud (map point cloud), respectively, describing line features in the target point cloud. θ and *t* denote the robot’s heading angle and two-dimensional position, and R(θ) represents a two-dimensional transformation matrix. *N* corresponds to the number of points in the source point cloud, and *i* represents the index within the source point cloud.

We employ the Gauss–Newton method to minimize the cost function detailed in Equation (Equation 1). This is translated into solving Equation (Equation 2), where x=[θ,t] represents the state variables.
(2)$$\delta x=-H^{-1}J^{T}e$$

In this context, we compute the Jacobian *J* of the residual function (Equation 1) with respect to *x* and the corresponding Hessian matrix $H=J^{T}J$. The vector $J^{T}e$ represents the projection component of the error *e* in the gradient direction and is denoted as $b=J^{T}e$.

Given that the Hessian matrix is positive definite and symmetric, eigenvalue decomposition and inversion are performed on H to derive its inverse.
(3)$$\begin{array}{cc} H^{-1} & =\begin{array}{cc} \left[{\vec{v}}_{1},{\vec{v}}_{2},{\vec{v}}_{3}\right] & \left[\begin{array}{ccc} \frac{1}{\lambda 1} & 0 & 0\\ 0 & \frac{1}{\lambda 2} & 0\\ 0 & 0 & \frac{1}{\lambda 3} \end{array}\right]{\left[\begin{array}{c} {\vec{v}}_{1},{\vec{v}}_{2},{\vec{v}}_{3} \end{array}\right]}^{-1} \end{array}\\ \; & =V\hat{\lambda }V^{-1} \end{array}$$

The eigenvectors of the Hessian matrix *H*, denoted as ${\vec{v}}_{1}$, ${\vec{v}}_{2}$, and ${\vec{v}}_{3}$, comprise the transformation matrix *V*. The corresponding eigenvalues are $\lambda_{1}$, $\lambda_{2}$, and $\lambda_{3}$, which form the diagonal elements of matrix $\hat{\lambda }$, the inverse of which represents the scaling of the vectors in the transformation. It follows that the update Equation (Equation 4) is given by
(4)$$\delta x=V\hat{\lambda }V^{-1}b,$$
where *b* is the error vector, and *V* signifies the transformation matrix constructed from the eigenvectors of the Hessian matrix. The diagonal matrix $\hat{\lambda }$, comprised of the eigenvalues’ reciprocals, indicates the scaling factors. The vector *b* resides within the vector space $X^{3}$. A transformation via *V* maps *b* into a new vector space $Y^{3}$, where it is scaled by $\hat{\lambda }$, then mapped back into the original space $X^{3}$ via $V^{-1}$.

Smaller eigenvalues, $\lambda_{i}$, amplify the transformation from *b* to δx, with an increased gain on the corresponding eigenvectors. This amplification causes δx to be more susceptible to instabilities due to perturbations along these directions.

In contrast, larger eigenvalues, $\lambda_{i}$, reduce the transformation magnitude from *b* to δx, resulting in smaller updates on the corresponding eigenvectors. This reduction contributes to the solutions’ stability along those directions.

We define the condition number, $\tau_{\lambda }=\lambda_{\max}/\lambda_{\min}$, to evaluate the matrix’s conditioning. A pre-set $\tau_{dg}$ is defined; exceeding this threshold indicates degeneration in the direction of the smaller eigenvalue.

#### 3.1.2. Contribution Sampling

When degeneration occurs, we calculate the contributions of different points to the optimization, focusing on the role of the translational elements in localization. In Figure 2, points in various locations present different normals after projection. The direction of these normals determines the constraint strength of the points. Points $p_{a}$ and $p_{d}$ apply the most substantial constraints on the vertical and horizontal directions, respectively. In contrast, due to a smaller angle with the laser, the normal at point $p_{b}$ imposes a more significant constraint than that of point $p_{e}$. Point $p_{c}$ imposes constraints in both the vertical and horizontal directions. Additionally, $v_{1}$ and $v_{2}$ represent the eigenvectors corresponding to the eigenvalues of the Hessian matrix, where a larger eigenvalue signifies a stronger overall constraint in its direction.

The contribution of a point to the optimization is proportionate to the point-to-line error it generates. This optimization contribution can be seen as the magnitude of the projection of the point-to-line error, after it undergoes a linear transformation, onto the optimization vector. This suggests that the magnitude of the point-to-line error and the direction of the normal collectively determine a point’s contribution during the optimization process.

Considering a 2D point $p_{i}\in R^{2}$, the point $p_{j_{1}^{i}}$ is a point in the line features of the target point cloud that define the normalized residual as follows:(5)$$\begin{array}{c} e_{i}={\vec{n}}_{i}^{T}\left(\frac{p_{i}-p_{j}^{1}}{|p_{i}-p_{j}^{1}|}\right) \end{array}$$

The next step involves calculating the Jacobian of the translational vector *t*:(6)$$\begin{array}{cc} J_{t}^{i} & =\frac{\partial (e)}{\partial t}\\ \; & ={\vec{n}}_{i}^{T} \end{array}$$

The error vector is defined as $b_{i}=J_{t}^{i}e_{i}$, referring to Equation (Equation 2). We introduce the contribution factor $\tau_{t}$ as follows:(7)$$\tau_{t}=V_{t}b_{i}$$

Here, $V_{t}$ is the matrix of eigenvectors associated with the translation segments in the Hessian matrix. The contribution factor $\tau_{t}$ represents the gain of the residual projection vector in the direction of the eigenvectors of the Hessian matrix.

Our analysis shows that directions corresponding to smaller eigenvalues are usually less restricted (Section 3.1.1). In the direction of eigenvectors corresponding to smaller eigenvalues, we calculate the contribution of each point. This $\tau_{t}$ value ranges from 0 to 1.

Point weights $w_{i}$ are defined as follows:(8)$$w_{i}=\left\{\begin{array}{cc} \eta , & \mathrm{if}\;(\tau_{t}>\tau_{ct}),\\ 1, & \mathrm{otherwise}. \end{array}\right.$$

Lesser importance is assigned a weight of 1. In contrast, surpassing threshold $\tau_{ct}$ is given a weight denoted by η, which is determined as follows:(9)$$\eta =\left(\frac{\lambda_{1}}{\lambda_{2}}\right)\cdot \frac{\tau_{t}}{\tau_{ct}}$$

In this context, the eigenvalues $\lambda_{1}$ and $\lambda_{2}$ of the Hessian matrix are associated with the translational components, with $\lambda_{1}$ being larger than $\lambda_{2}$. The objective is to give greater weights to points with higher τ values.

#### 3.1.3. Point-to-Line ICP

We iterate to compute the weights $w_{i}$ for the current point cloud frame. For indoor environments with more line features, we introduce a new point-to-line optimization equation incorporating these weights, as illustrated in Equation (Equation 10). Unlike [9], we assign greater weights to points with higher contributions. In this equation, our goal is to minimize the error between the point cloud and the map through iterative optimization, thereby searching for the optimal heading angle θ and translation *t*.
(10)$$\min_{\theta ,t}\sum_{i=1}^{N}∥w_{i}{\vec{n}}_{i}^{T}\left[R\left(\theta \right)p_{i}+t_{k+1}-p_{j_{1}^{i}}\right]∥$$

In this optimization function, $w_{i}$ represents the weight corresponding to each point, while the meanings of the other variables remain consistent with Equation (Equation 1). Through this optimization equation, points that significantly contribute to the degradation direction are assigned higher weights. During the optimization process, these points impose greater constraints, essentially improving the structure of the Hessian matrix and resulting in a more reasonable and uniform distribution of eigenvalues. This helps limit the drift phenomenon in the positioning results.

### 3.2. Back-End Map Monitoring Module

#### 3.2.1. Ray-Casting Calculation

We utilize an advanced ray-casting algorithm, as originally proposed in [26], to detect static anomalies. It is widely used for environment mapping in laser SLAM systems, and it projects laser beams and records where they intersect with surroundings, as detailed in [27,28]. Our study adapts this approach to use a grid-based map for efficient anomaly detection.

In the process of ray-casting, each grid’s location along the laser trajectory is checked. If a beam’s impact point coincides with a free grid cell (unoccupied) and there are not any occupied grid cells in the map within the pre-set $\tau_{dis}$, the cell in question is considered a positive object and assigned a value of 1. This indicates the prospective presence of a positive object in that location. In contrast, if the laser beam intersects with occupied grid cells on the map, and no free grid cells within the preset distance threshold $\tau_{dis}$ exist, these cells are identified as negative objects and are indicated by a value of 0. This method effectively filters out spurious results that are often attributed to sensor noise or environmental proximity effects. A demonstration of the ray-casting-based map change detection method is shown in Figure 3a–d, and it is calculated according to Equation (Equation 11).
(11)$$D_{i,j}=RayCasting\left(M_{i,j}\right)$$

Here, $M_{i,j}$ represents the original grid map, and $D_{i,j}$ represents the map detection result, which includes the detected changed states of the grid cells. The indices *i* and *j* correspond to the row and column indices of the grid, respectively, indicating the detected changed grid states.

#### 3.2.2. Multi-View Voting

To enhance detection accuracy and minimize false positives, this study proposes a detection method utilizing the Multi-View Weighted Voting mechanism. This mechanism leverages the combined observations of map anomalies made by the robot from various poses (see Figure 4).

The initial states of the map’s grid cells can be denoted as $M_{i,j}^{0}$, where *i* and *j* represent the row and column indices of the grid cells, respectively. Assuming that the robot acquires laser data from *N* different viewpoints denoted as $L^{1},L^{2},...,L^{N}$, each viewpoint’s pose is represented as $T^{k}$, where *k* denotes the *k*-th viewpoint. The detection results of the laser scan data under pose $T^{k}$ are denoted as $D_{i,j}^{k}$, indicating the state detection results of grid cell (i,j) under the *k*-th viewpoint.

We define a voting function *V* that takes the state detection results of grid cell (i,j) from *N* viewpoints as input and produces the outcome of the multi-view voting. The pre-set parameter $\beta^{m}$, ranging between 1/2 and 1, represents the truncation coefficient. This coefficient establishes the threshold for the acceptance of polling outcomes; a higher value signifies a greater proportion of concordant votes required for an outcome to be validated, whereas failure to meet this threshold results in the rejection of the outcome.

More specifically, when the weighted result of detections from multiple viewpoints exceeds the threshold $\beta^{m}N$, it indicates high consistency in positive detections. Conversely, when the weighted result falls below the threshold $(1-\beta^{m})N$, it indicates high consistency in negative detections. If the weighted result falls between these thresholds, it suggests low consistency in detections from multiple viewpoints, leading to low confidence. As a result, the detection is discarded, and the map position remains unchanged.
(12)$$V\left(D_{i,j}^{1},D_{i,j}^{2},...,D_{i,j}^{N}\right)=\left\{\begin{array}{cc} 1, & \mathrm{if}\;\sum_{k=1}^{N}w_{i,j}^{k}\cdot D_{i,j}^{k}>\beta^{m}N\\ 0, & \mathrm{if}\;\sum_{k=1}^{N}w_{i,j}^{k}\cdot D_{i,j}^{k}<\left(1-\beta^{m}\right)N\\ M_{i,j}^{0}, & \mathrm{otherwise} \end{array}\right.$$

Here, $w_{i,j}^{k}$ denotes the weight assigned to the grid position corresponding to viewpoint *k*. As the laser beams are subject to measurement errors, these errors become more pronounced as the measurement distance increases. Consequently, it is desirable to assign higher weights to laser points that are closer in proximity relative to those that are farther away.

Assuming that the laser has a maximum range of $R_{\max}$ and a minimum range of $R_{\min}$, the distance from the grid cell (i,j) to the robot under pose *N* can be represented as $d_{i,j}^{k}$. Furthermore, let $\alpha^{m}$ denote the weight adjustment hyperparameter. The weight $w_{i,j}^{k}$ can be calculated using the following expression:(13)$$w_{i,j}^{k}=\left\{\begin{array}{cc} 1-\alpha^{m}\left(\frac{d_{i,j}^{k}-R_{\min}}{R_{\max}-R_{\min}}\right), & \mathrm{if}\;d_{\min}\leq d_{i,j}^{k}\leq d_{\max}\\ 0, & \mathrm{otherwise} \end{array}\right.$$

In the weight definition of the above equation, to avoid errors introduced by laser blind spots and extremely long distances, we define the effective range of the laser beam as $d_{\min}$ and $d_{\max}$. Additionally, considering the error characteristics of laser ranging, we set the weight of measurements closer within the range to be smaller, while those farther away have larger weights.

#### 3.2.3. Map Update

By voting on the state detection results from all viewpoints, the final detection result of grid map changes $D_{i,j}^{\mathrm{final}}$ can be obtained.
(14)$$D_{i,j}^{\mathrm{final}}=V\left(D_{i,j}^{1},D_{i,j}^{2},...,D_{i,j}^{N}\right)$$

Here, $D_{i,j}^{\mathrm{final}}$ represents the portion of the map that has changed. Updating $M_{i,j}^{0}$ based on this result yields the final map $M_{i,j}$. The updated grid map $M_{i,j}$ is sampled as a point cloud map and provided to the front-end module for registration.

After the map-updating process described above, the robot enters the changed environment and can detect the areas of change in real time. based on the voting mechanism, the map is updated accordingly. The front-end localization module can then perform point cloud map registration more robustly, providing stable pose information as output.

## 4. Results

We performed experiments in specific and comprehensive scenarios using the HeiYi unmanned vehicle (Tufeng Technology Co., Ltd., Beijing, China), equipped with an array of sensors: Siminics Pavo2 (Siminics Optoelectronics Technology Co., Ltd., Shanghai, China) single-line LiDAR with a 270° field of view and 50 m range, CLU3.0 MEMS IMU (Tufeng Technology Co., Ltd., Beijing, China), Livox MID360 multi-line LiDAR (DJI Technology Co., Ltd., Shenzhen, China), Daheng industrial camera, wheel speedometer, and OptiTrack markers (from VIRTUAL POINT Co., Ltd., Beijing, China) (Figure 5). The Pavo2 LiDAR features an angular resolution of 0.08°.

For the ground truth, we employed the Goslam T300 Pro (Tianqing zhizao Aviation Technology Co., Ltd., Beijing, China) handheld scanner for high-precision point clouds, the MID360 LiDAR for accurate position registration, and the OptiTrack motion capture system for supplementary truth data.

### 4.1. Simulation Study

To evaluate the effectiveness of the proposed algorithm in degradation environments, we simulated a long corridor using Matlab 2018b, as shown in Figure 6a. The corridor walls are on either side, with the laser sensor positioned on the left. Vertical obstacles of varying lengths were placed in the middle to simulate the limited constraint points in a long corridor environment. In the experiment depicted in Figure 6b, we compared the horizontal axis localization accuracy of the CSMUL algorithm and the ICP matching algorithm with different numbers of reflection laser points. We set an initial position error of −0.5 m, 0.3 m for the calculation. The results indicate that due to the enhanced weighting of obstacle laser reflection points in the CSMUL algorithm (Equation (Equation 10)), the algorithm maintains centimeter-level error even with a few points, and the error decreases further as the number of points increases. In contrast, the ICP algorithm fails to maintain high accuracy with few reflection points, resulting in an overall greater error than that of CSMUL.

Furthermore, we analyzed three sets of comparative experiments with different numbers of laser reflection points to examine the convergence of the CSMUL and ICP algorithms. Figure 7 shows the incremental changes in the x direction with different numbers of obstacle reflection points. The increment in x reflects the algorithm’s ability to converge from the initial value to the optimal solution. It can be seen that with two, four, and six laser reflection points from the middle obstacles and with more than 400 laser points from the corridor walls, the CSMUL algorithm initially shows significant x-direction increments, which gradually decrease to near zero as the number of iterations increases.

In contrast, the x-direction increments in the ICP algorithm remain consistently small and quickly approach zero, preventing the x-direction calculations from iterating to the optimal value and resulting in horizontal degradation. In this experiment, the contribution sampling mechanism in the CSMUL algorithm enhances the weighting of laser reflection points from obstacles in the simulation, thereby increasing the robustness of the localization system.

### 4.2. Efficiency Study

The efficiency of CSMUL is evaluated by computing the average processing time of each step on the RK3588s (https://www.rock-chips.com (accessed on 10 June 2024)) SoC, which is a typical ARM-based embedded platform. We conducted tests in complex office map scenarios, with approximately 1800 points per frame and a point cloud input frequency of 10 Hz. The primary testing modules of the CSMUL algorithm include front-end point cloud processing with PL-ICP, degradation detection, contribution sampling, and total front-end processing time. The back-end includes map detection and map updates.

Figure 8a shows the computation times for the front-end and back-end modules. The average times for the front-end and back-end are 36 ms and 11 ms, respectively. Overall, the processing time is less than the laser input interval, ensuring good real-time performance. As shown in Figure 8b, the front-end modules with the highest processing time are PL-ICP and contribution sampling due to the need to traverse the point cloud and perform Jacobian calculations. Degradation detection consumes the least time because the eigenvalue decomposition of the Hessian matrix is only performed once. Figure 8c illustrates that the back-end modules related to map processing involve real-time ray-casting detection and updates during point cloud traversal, which do not require extensive floating-point calculations, resulting in minimal time consumption.

Additionally, it is worth mentioning that the front-end and back-end modules run in parallel in two independent processes, communicating via ROS topics.

### 4.3. Degeneration Environment

We assessed the CSMUL algorithm’s localization performance in a degraded environment by conducting tests in a 40 m corridor, as shown in Figure 9. The robot traversed the area, mainly capturing laser point clouds on the side walls with few forward constraints. We implemented the proposed CSMUL algorithm for comparative evaluation with Adaptive Monte Carlo Localization (AMCL).

AMCL is a widely used real-time localization algorithm for mobile robots that employs a particle filter to estimate the robot’s pose within a map. Known for its self-adaptation, robustness, and accuracy, it is one of the mainstream algorithms in the industry. Therefore, choosing AMCL for comparison effectively represents the efficacy of our improved method.

The pre-set hyperparameters used for this and all subsequent experiments are summarized in Table 1. $\tau_{dg}$ represents the threshold for degradation detection, with larger values indicating more severe degradation; $\tau_{ct}$ represents the threshold for contribution determination, with smaller values considering more contribution points; $\tau_{dis}$ is the limit distance for ray-casting and is set according to the radar’s measuring range; $\beta^{m}$ represents the confidence level of multi-view voting; $\alpha^{m}$ is mainly used to adjust the confidence of different distance measurements in map updates.

Figure 9b,c show the two algorithms’ performance in the corridor. AMCL suffered from the corridor’s smooth walls and feature scarcity, as is evident from its map mismatch in Figure 9c. In contrast, our CSMUL algorithm applied contribution-based sampling to focus on key points, effectively preventing degeneration, as demonstrated in Figure 9b.

Figure 10 compares the trajectory and positioning errors between the algorithms. Table 2 shows the statistical errors. Our CSMUL algorithm closely matches the reference trajectory with minimal errors, even in this feature-less corridor, whereas AMCL encounters over four meters of drift at its peak, as depicted in Figure 9b,c.

### 4.4. Map Evolution Environment

We established an environment in an optical motion capture laboratory, as depicted in Figure 11a. Various obstacles of different sizes and spacing were placed within the camera’s field of view, as shown in Figure 11b.

To begin, we generated a reference point cloud map using a scanner, which was then projected as a 2D map for the comparison of localization algorithms. Next, we randomly repositioned the obstacles to alter the map environment, as illustrated in Figure 11c, in order to simulate variations in both the map environment and obstacle movement in a real-world setting.

After repositioning the obstacles and modifying the layout, we executed the localization algorithm with map anomaly detection and updating. The truncation coefficient $\beta^{m}$ and weight adjustment coefficient $\alpha^{m}$ were set to 2/3 and 0.2, respectively. Figure 12 presents several steps of this process. In Figure 12a, potential locations of anomalies were determined using ray-tracing, indicating areas where the map environment may have changed due to obstacle movement. Figure 12b displays the detected map anomalies in yellow, while the green regions represent the relocated obstacles. Figure 12c depicts the original map state prior to the obstacle movement, and Figure 12d shows the updated map where the central obstacle was moved to the bottom right, resulting in a change in its relative position.

The objective of map change detection and updating is to enhance the robustness of the localization algorithm. Meanwhile, the localization algorithm itself supports map change detection and updating. In the map change localization experiment, we utilized the AMCL algorithm as a benchmark and employed the pose information provided by the optical motion capture system as the reference trajectory.

Figure 13 illustrates the experimental results. The green trajectory represents the reference trajectory, while the red trajectory depicts the significant drift and localization failure of the AMCL algorithm when it is near obstacles and the field of view is obstructed. In contrast, our algorithm can provide relatively stable and accurate pose information in varying environments. Table 3 shows the statistical errors.

### 4.5. Complex Integrated Environment

To investigate the impact of robustness measures on localization performance, experiments were conducted in a diverse environment encompassing various areas, including long corridors, office spaces, laboratories, and warehouses (refer to Figure 14). Accurate three-dimensional point cloud maps were acquired using a 3D scanner, serving as the ground truth for evaluation.

Within this complex environment, the laser localization algorithm encounters several challenges, such as limited local space and the presence of map objects that are susceptible to variations, such as chairs and temporary obstacles in office spaces. To evaluate the impact of the proposed enhancements on the localization accuracy in a representative scenario, a trajectory route (depicted in Figure 15) was selected, encompassing most areas of the entire floor. Examination of Figure 15 reveals scene and trajectory errors, highlighting significant drift problems in single-line laser localization caused by object movement and spatial constraints in complex office areas. The maximum error observed with the AMCL algorithm exceeded 3 m, whereas the proposed CSMUL algorithm achieved a maximum error of less than 0.6 m.

Figure 16 presents a comparison of the position and orientation errors, demonstrating the superior error control of the CSMUL algorithm in contrast to AMCL. The cumulative distribution function (CDF) of errors, illustrated in Figure 17, clearly indicates the advantages of the CSMUL algorithm over AMCL with regard to both position and orientation errors.

## 5. Conclusions

This paper presents a robust localization framework for complicated environments using a single-line LiDAR. The framework combines degradation detection, contribution sampling, and map anomaly detection and updating techniques to achieve stable and optimized alignment positioning in complex environments. This significantly enhances the robustness and performance of the positioning system. However, it is important to note that the methods discussed in this paper are currently applicable only to 2D environments, and extensions are necessary to apply them to localization and navigation in 3D environments.

In our study, we employed several crucial threshold parameters. The proper setting of these parameters is crucial for achieving more desirable results. Despite the limitation of setting these parameters manually, the performance and adaptability of the framework can be further enhanced in the future through adaptive parameter tuning, cross-sensor fusion localization techniques, and data-driven approaches.

The research methodology presented in this paper introduces valuable ideas in the fields of 2D LiDAR localization, laser SLAM, laser odometer stability improvement, and robot repositioning. This is significant for providing more reliable and accurate localization solutions in areas such as robot navigation, unmanned vehicles, and intelligent transportation systems. Further research could explore how this framework can be integrated with other sensing and localization technologies to further improve the positioning performance and robustness of the system.

## Figures and Tables

**Figure 1 sensors-24-03927-f001:**
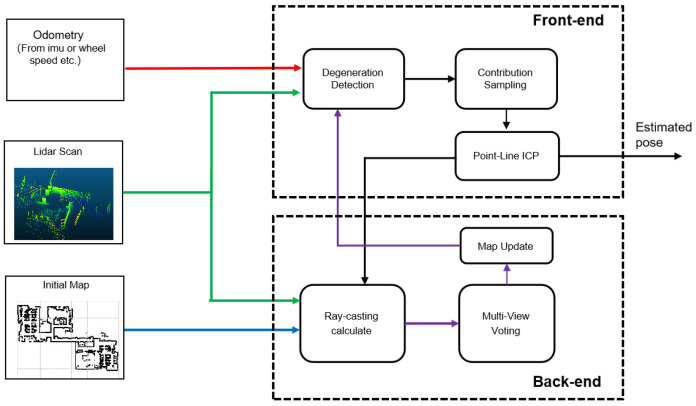
Main diagram of CSMUL.

**Figure 2 sensors-24-03927-f002:**
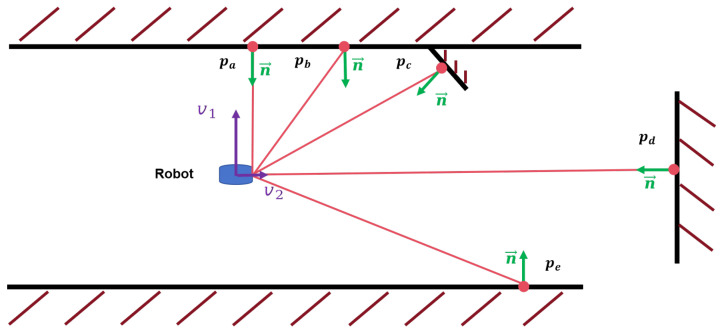
A diagrammatic representation of point position constraints for various locations within a specific scenario for laser localization.

**Figure 3 sensors-24-03927-f003:**
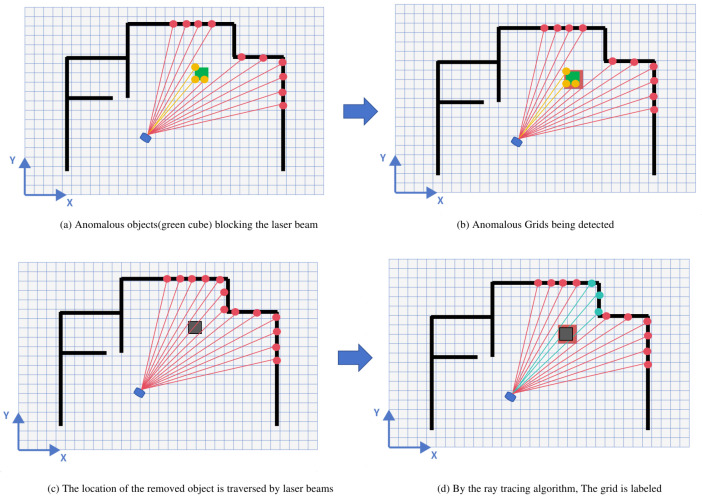
Map anomaly detection based on ray-casting.

**Figure 4 sensors-24-03927-f004:**
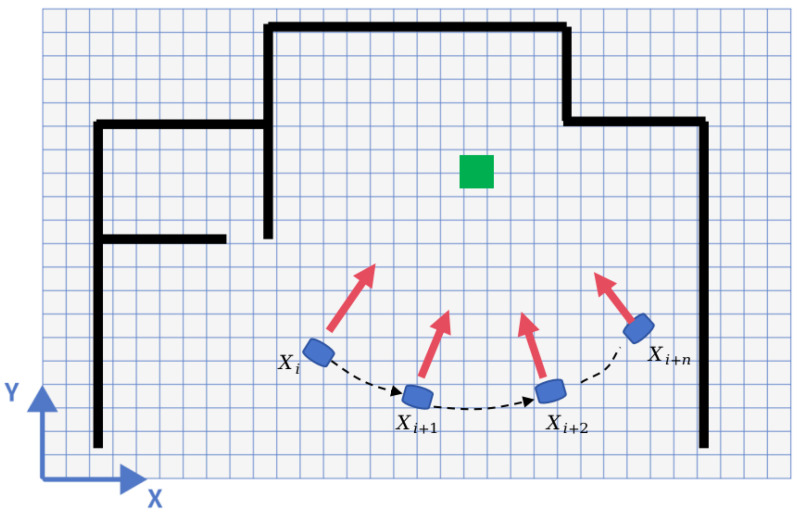
Multi-View Voting. The green region represents map change points, and the detection status is determined based on the collective observations from multiple poses.

**Figure 5 sensors-24-03927-f005:**
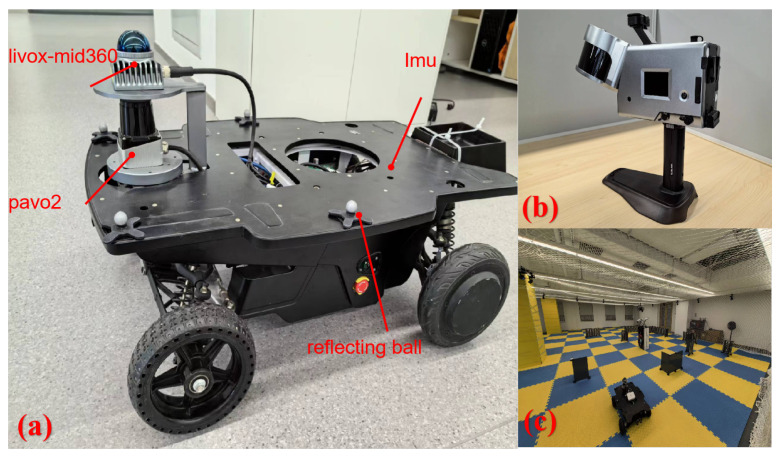
Overview of the experimental platform: (**a**) robot vehicle and integrated sensors; (**b**) Goslam scanner used as the reference ground-truth system; (**c**) OptiTrack motion capture system used as the reference ground-truth system.

**Figure 6 sensors-24-03927-f006:**
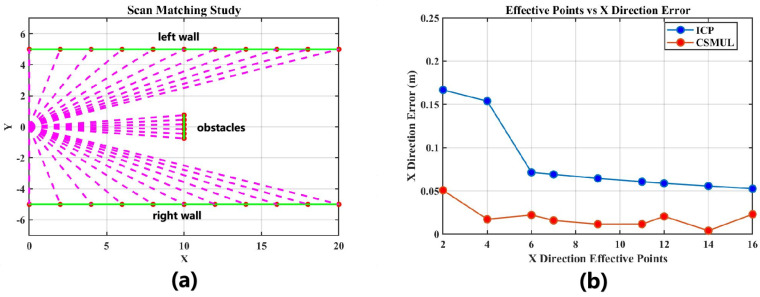
Simulation environment and error evaluation. (**a**) Long corridor simulation environment in Matlab, including walls on both sides and middle obstacles and laser beams. (**b**) Comparison of the horizontal x−direction positional error between the CSMUL and ICP algorithms for different numbers of reflection points.

**Figure 7 sensors-24-03927-f007:**
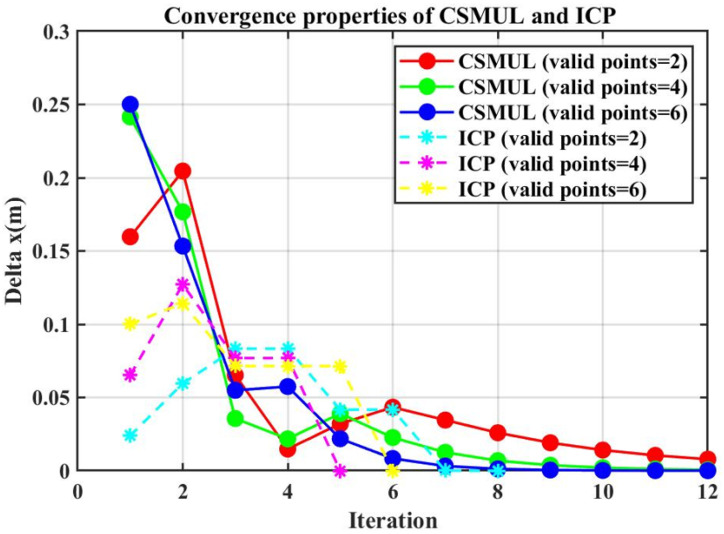
Study of convergence properties: The x-axis represents the number of iterations, and the y-axis shows the incremental changes in the horizontal direction (x direction in the simulation) across different algorithm iterations. The CSMUL algorithm demonstrates strong convergence in the horizontal x direction, whereas the ICP algorithm shows minimal gain and fails to converge, resulting in degradation.

**Figure 8 sensors-24-03927-f008:**
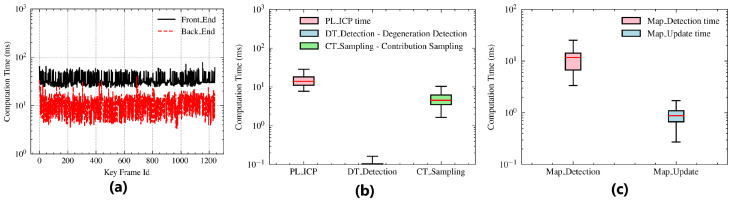
Computation time of the CSMUL algorithm. (**a**) Front- and back-end computation time. (**b**) Computation time of each module in the front-end. (**c**) Computation time of each module in the back-end.

**Figure 9 sensors-24-03927-f009:**
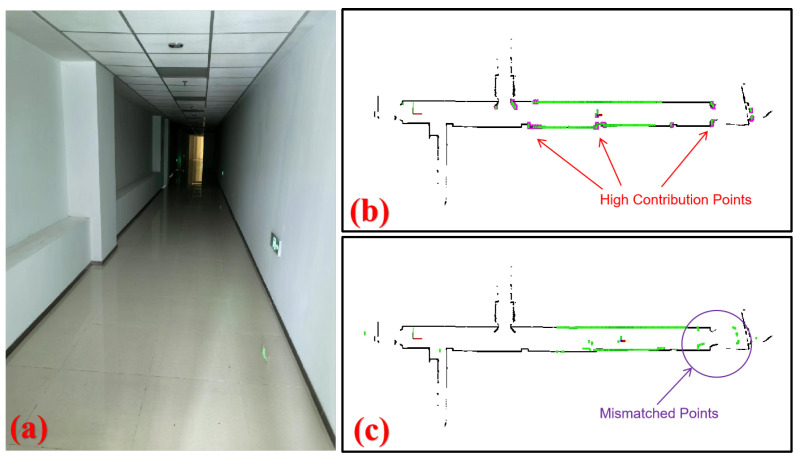
Algorithm comparison in a degraded corridor. (**a**) Experimental setup. (**b**) Our CSMUL algorithm avoids localization failure by identifying high-contribution points (pink color), and the laser scan (green color) matches well with the map (black color). (**c**) AMCL suffers from significant drift, and the laser scan completely fails to match.

**Figure 10 sensors-24-03927-f010:**
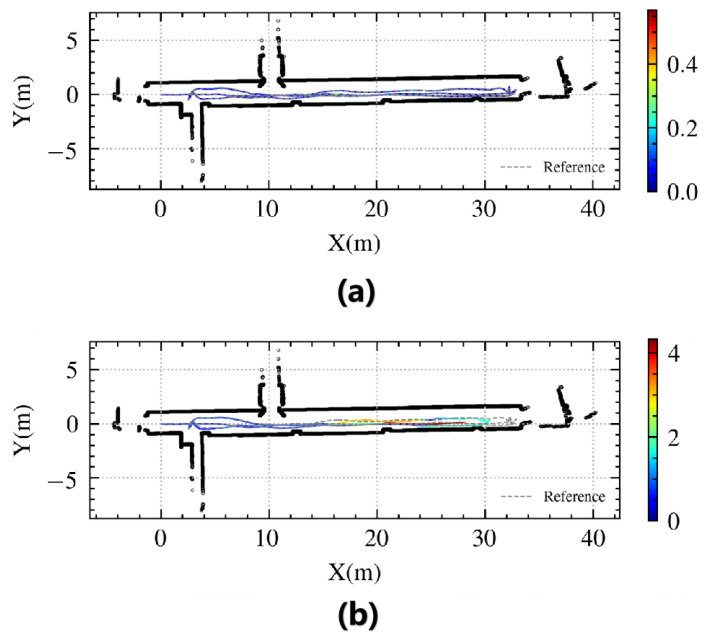
Trajectory comparison in a degraded corridor (the colorful lines indicate the algorithm’s path, with the gray dashed lines as a reference): (**a**) CSMUL exhibits minimal positional error in its trajectory; (**b**) AMCL suffers noticeable drift in its trajectory.

**Figure 11 sensors-24-03927-f011:**
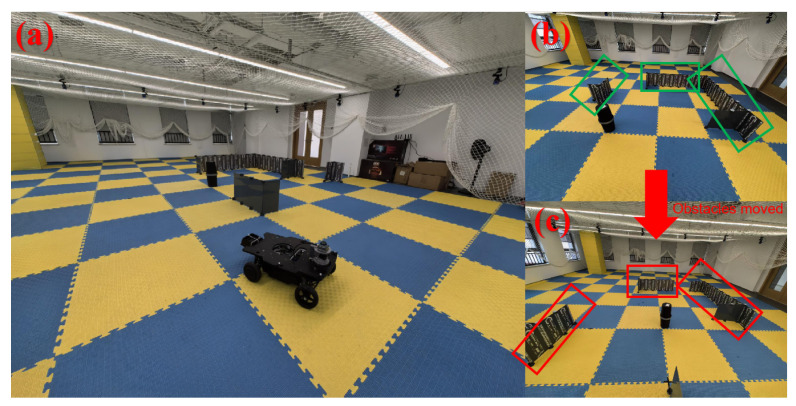
Experimental setup for map changes based on optical motion capture. (**a**) Layout of the laboratory site with pose tracking conducted by 20 high-speed cameras. (**b**) Placement of obstacles in the scene, forming the original map. (**c**) Movement of obstacles shown in (**b**), resulting in changes in the map layout.

**Figure 12 sensors-24-03927-f012:**
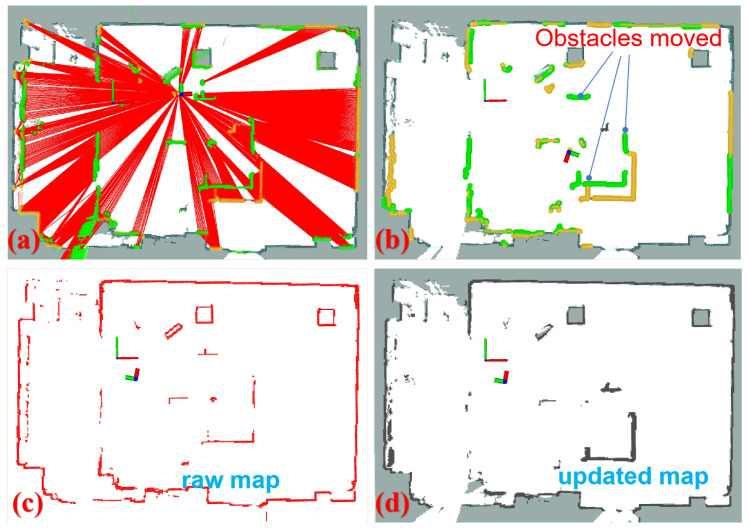
Experiment on map anomaly detection and updating. (**a**) Ray-casting is employed to detect anomalies. The red virtual rays indicate potential areas of map changes, while the regions not covered in red represent areas that remain unchanged according to the ray-tracing computation. (**b**) The detected anomalies are highlighted in yellow, and the green regions represent the relocated obstacles. (**c**) The original map. (**d**) The updated map.

**Figure 13 sensors-24-03927-f013:**
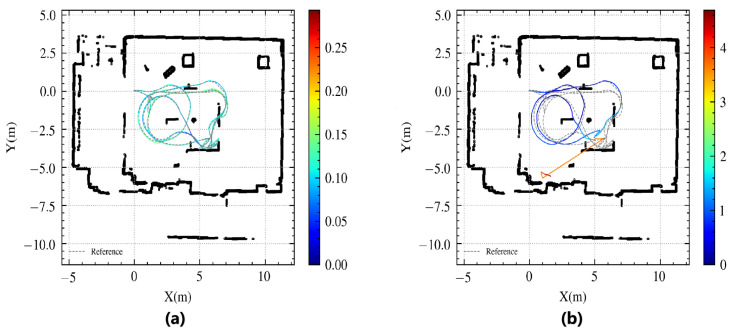
Evaluation of localization trajectories under map change conditions (the colorful lines indicate the algorithm’s path, with the gray dashed lines as a reference). (**a**) CSMUL (proposed). (**b**) AMCL.

**Figure 14 sensors-24-03927-f014:**
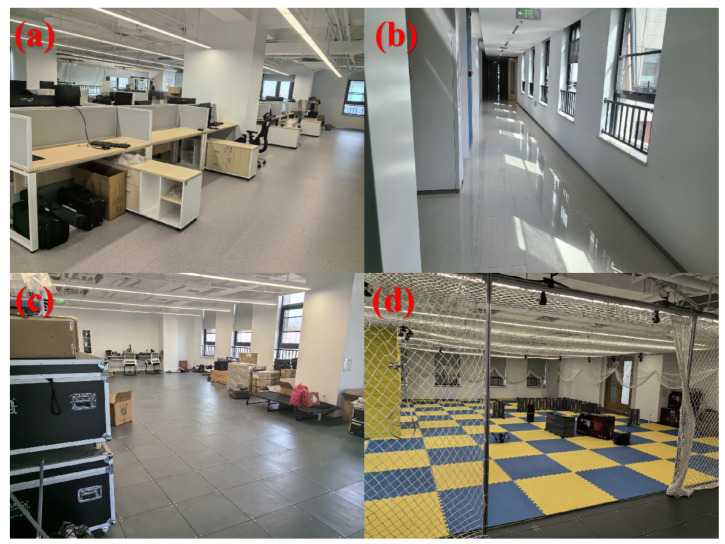
Integrated experimental scenes in an office building. (**a**) Office area. (**b**) Corridor area. (**c**) Warehouse scene. (**d**) Laboratory scene.

**Figure 15 sensors-24-03927-f015:**
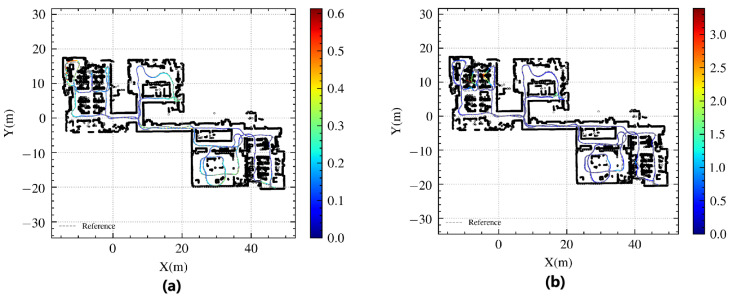
Evaluation of localization trajectories in complex integrated environment (the colorful lines indicate the algorithm’s path, with the gray dashed lines as a reference). (**a**) CSMUL (Proposed). (**b**) AMCL.

**Figure 16 sensors-24-03927-f016:**
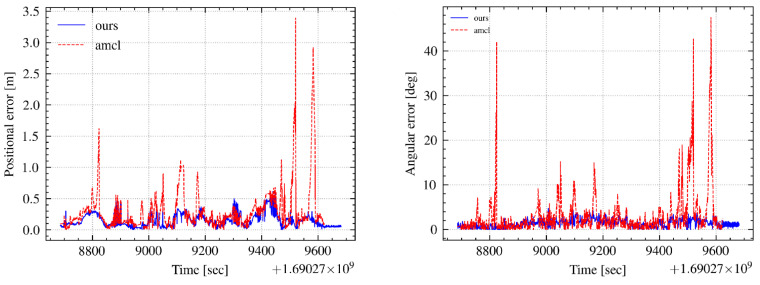
Pose error assessment in complex integrated scenarios.

**Figure 17 sensors-24-03927-f017:**
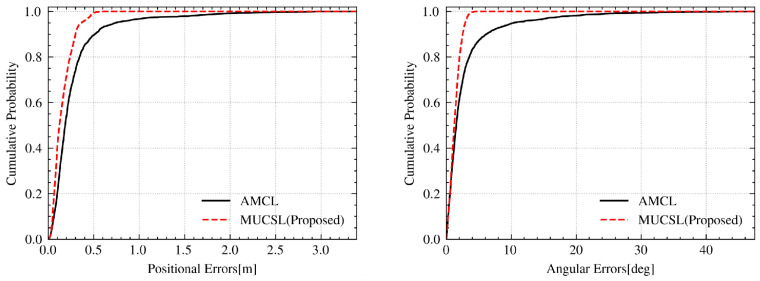
CDF evaluation in complex integrated scenarios.

**Table 1 sensors-24-03927-t001:** Hyperparameter settings in the experiment.

$\tau_{\mathrm{dg}}$	$\tau_{\mathrm{ct}}$	$\tau_{\mathrm{dis}}$	$\beta^{m}$	$\alpha^{m}$
3.0	0.65	2.0	2/3	0.2

**Table 2 sensors-24-03927-t002:** Performance metrics from the narrow corridor experiment.

Algorithm	Maximum Error (m)	Mean Error (m)	RMSE (m)
CSMUL (Proposed)	0.515	0.092	0.111
AMCL	failed ^1^	1.292	1.931

^1^ AMCL exhibited large localization drifts in this experiment.

**Table 3 sensors-24-03927-t003:** Performance metrics from the map change experiment.

Algorithm	Maximum Error (m)	Mean Error (m)	RMSE (m)
CSMUL (Proposed)	0.293	0.112	0.114
AMCL	failed ^1^	0.911	1.266

^1^ AMCL exhibited large localization drifts in this experiment.

## Data Availability

The raw data supporting the conclusions of this article will be made available by the authors upon request.

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
