# Peer review of "Single-Line LiDAR Localization via Contribution Sampling and Map Update Technology"

_sensors, 2024, doi:10.3390/s24123927_

Round 1
Reviewer 1 Report
Comments and Suggestions for Authors
The authors have proposed the CSMUL algorithm that incorporates weighted contribution sampling and dynamic map updating to address scene degeneracy and frequent map changes problems. Some advice can be taken into consideration.
1. the Gauss-Newton method is not the new thing to solve the optimization problem. Why did the author use the method to be the potential solution? Please give some necessary explanations.
2. Why did the authors use Ray-casting algorithm?
3. Please give more surveys of the related topics within three years. The current citation is to some extent out of date.
4. There are some typos and English grammar issues. Please modify the paper.
5. Honestly, the work just combines some proposed method into one framework. The novelty is limited. Luckly, the experiments can sufficiently support the idea. It would be better if some comparisons can be made in the simulation.
6. It would be better if the authors make discussion instead of directly demonstrating the outperformance of the proposed method.
Comments on the Quality of English LanguageExtensive editing of English language required
Reviewer 2 Report
Comments and Suggestions for Authors
Please see the attached file.

Good.
Reviewer 3 Report
Comments and Suggestions for Authors
The authors propose a Contribution Sampling and Map Updating Localization (CSMUL) algorithm for a single-line LiDAR localization. The algorithm incorporates techniques of weighted contribution sampling and dynamic map updating. The topic is very challenging and it is fully in the scope of the journal. The presented results are sufficient and well organized, which helps the reader to understand the contribution of the submission.
Some remarks and suggestions:
1. There are some punctuation errors such as missing space between characters (mainly before parenthesis – lines 18-22, 75, 122, 150, 209 and some more).
2. The 2-norm symbol in (1) and (10) must be revised as .
3. Maybe the referenced equation in line 158 is (2) not (4)?!
4. There is no reference to equation (11). It is not introduced in the text.
5. Figure numbers must be added into the text between lines 282-286.
6. Adding a legend to Figures 5, 6, 7 could be considered. There are lines in different colors, which are not explained in the text (for example what is denoted with green line in Figure 6?)
7. It would be useful for the interested researchers to know much for the experience that the authors have in setting the threshold parameters of the proposed algorithm. Any directions would be useful for those who wanted to implement the CSMUL algorithm.
Round 2
Reviewer 1 Report
Comments and Suggestions for Authors
No further comments.
Comments on the Quality of English LanguageMinor editing of English language required
Reviewer 2 Report
Comments and Suggestions for Authors
I see that the authors have addressed the technical comments I provided previously. I belive the revised version is now ready for publication.
